# Stability of the standard incus coupling of the Carina middle ear actuator after 1.5T MRI

**Guy Fierens**[1]*, **Nicolas Verhaert**[1,2], **Farida Benoudiba**[3], **Marie-France Bellin**[4,5], **Dennis Ducreux**[3,4], **Jean-François Papon**[4,6], **Jérôme Nevoux**[4,6]

**1** ExpORL Research group, Department of Neurosciences, KU Leuven - University of Leuven, Leuven, Belgium, **2** Department of Otolaryngology, Head and Neck Surgery, University Hospitals Leuven, Leuven, Belgium, **3** AP-HP, Hôpital Bicêtre, Service de Neuroradiologie, Le Kremlin-Bicêtre, France, **4** Université Paris-Saclay, Faculté de Médecine Université Paris-Sud, Le Kremlin-Bicêtre, France, **5** AP-HP, Hôpital Bicêtre, Service de Radiologie Générale, Le Kremlin-Bicêtre, France, **6** AP-HP, Hôpital Bicêtre, Service d'Oto-Rhino-Laryngologie, Le Kremlin-Bicêtre, France

* guy.fierens@kuleuven.be

**Data Availability Statement:** All relevant data are within the paper and its Supporting Information files.

## Abstract

Limited data is available concerning the safety of active middle ear implants (AMEI) during Magnetic Resonance Imaging (MRI). Measurements in temporal bones are the gold standard for preclinical assessment of device safety. In this study the coupling stability of an actuator as used in a fully implantable AMEI was determined in temporal bones. Eleven temporal bones were implanted with the actuator according to the manufacturer's surgical guidelines. The actuator was coupled on the incus short process as recommended for sensorineural hearing loss. Temporal bones were exposed 10 times to the MRI magnetic field by entering the MRI suite in a clinically relevant way. Computed Tomography (CT) images were acquired before and after the experiment to investigate the risk of actuator dislocation. Based on the electrical impedance of the actuator, the loading of the actuator to the incus was confirmed. Relative actuator displacement was determined on the CT images by comparing the initial with the final actuator position in 3D space. Impedance curves were analyzed after each exposure to check the loading of the actuator to the ossicles. Analysis of CT images with a 0.30.6 mm in-plane resolution indicate no actuator displacement. The maximum detected change in impedance for all actuators was 8.43 Ω at the actuator's resonance frequency. Impedance curves measured when the actuator was retracted from the short process after the experiment still indicate the presence of a clear resonance peak. No actuator displacement or dislocation could be detected in the analysis of CT images and the measured impedance curves. Impedance curves obtained when the actuator was retracted from the incus short process still show a clear resonance peak, indicating the device is still functional after the MRI exposures.

## Introduction

Over the last decades, a number of different active middle ear implants (AMEI) have shown a succesful outcome in patients with different types of pathologies and hearing losses. The Carina® implant (Cochlear Ltd., Sydney, Australia) is the only fully-implantable AMEI for

**Funding:** The investigator-initiated research was partially funded by Cochlear Limited Research and Development under Grant Number IIR-2028. NV is partially supported by Research Foundation Flander, GF is employed by Cochlear Ltd and partially funded by Flanders Innovation and Entrepreneurship (HBC.2018.0184). GF participated in this study as a PhD researcher at the university of Leuven, Belgium. Cochlear Ltd had no further role in study design, data collection and analysis, decision to publish, or preparation of the manuscript.

**Competing interests:** I have read the journal's policy and the authors of this manuscript have the following competing interests: GF is an employee of Cochlear Ltd. This does not alter our adherence to PLOS ONE policies on sharing data and materials.

patients with moderate to severe sensorineural, conductive or mixed hearing loss currently on the market [1]. The Carina AMEI features a sensitive implantable subcutaneous microphone to pick up sound which is amplified in the implant body and converted into mechanical vibrations by the implantable actuator to stimulate a patient. Depending on a patient's specific needs, the actuator can be coupled to the ossicles, the oval window or the round window to compensate the hearing loss [2]. Coupling the actuator to the body of the incus has been used most in practice as it does not require mounting an additional prosthesis to the actuator tip [1, 3–5].

In addition to an increasing amount of people being implanted, Magnetic Resonance Imaging (MRI) is established as an important imaging technique in many clinical situations [6]. For patients with implantable devices however, getting an MRI scan is not straightforward due to safety concerns [6, 7]. One of the concerns are magnetically-induced forces and torques on magnetic components in the implant [8], which could lead to the displacement of the component resulting in patient harm. Most current hearing implants feature an implantable cylindrical Neodymium magnet which provides alignment between the implant and an externally worn sound processor to allow a transcutaneous, inductive coupling between both components. Magnetically-induced forces and/or torques acting on these magnets have proven to cause discomfort to patients, and in rare cases dislodge the magnet from its silicone housing, creating trauma and pain and an urgent need for revision surgery [9, 10]. Additionally, magnet dislodgement may cause adverse biological events when it goes undetected and untreated such as infections in the skin region surrounding the implant site, transdermal magnet extrusions or meningitis as a result of an infection spreading along the implant [10]. Actuators of AMEI like the Carina implant feature an electromagnetic actuator. Similar magnetically-induced forces could lead to a displacement of the actuator which could result in dislocation of the actuator from the ossicles or even damage the ossicular chain itself. Up until now, the Carina implant is labeled MRI-unsafe, indicating that a patient with this device can never undergo an MRI scan. The only MRI-conditional AMEI on the market at the moment is the Vibrant Soundbridge® 503 implant (MED-EL, Innsbruck, Austria) [11], which can be used for field strengths of 1.5 Tesla (T). In a prior version of this device, several issues have been detected related to the device's electromagnetic actuator [12]. Studies by Todt et al. [12, 13] indicated that the implanted actuator could move due to magnetically-induced forces, leading to a number of actuator dislocations. Patients have also reported tremendous pain in the middle ear when being scanned with this device, leading to a premature ending of the procedure [12]. Additional experiments in temporal bones by Jesacher et al. [14] however indicated that the induced forces would not lead to irreversible damage to a patient's ossicular chain.

For the Carina implant, knowledge is lacking on the stability of the actuator when being exposed to a 1.5 T magnetic field. The present study will therefore investigate the risk of dislocating the actuator from its standard fixation on the incus body when exposed to a 1.5T magnetic field. The loading of the actuator will be tested before and after exposure to a 1.5 T field to assess the coupling efficiency. An actuator will be rigorously coupled to the body of the incus in order to represent the most commonly used configuration [1, 3–5]. The coupling stability is observed in the case when a patient traverses the fringe field of a 1.5 T MRI scanner in a clinically relevant manner. Magnetically-induced forces are directly proportional to magnetic field gradients [8, 15], which are considerably large when a patient crosses the scanner's fringe field before or after an examination. After crossing the fringe field, the patient is put in the isocenter of the scanner. In this location the homogeneity of the field is maximal, and the induced torque will be maximal as well [8].

## Materials and methods

Data was collected from eleven ears in fresh-frozen, anonymized human specimen obtained from L'école de Chirurgie du Fer à Moulin (Paris, France). Temporal bones (TBs) were harvested and used after the surgical procedure in accordance with the Helsinki declaration and approved by the ethics committee of Le Bicétre hospital (Paris, France) in agreement with article R2213-13 of the "Code Général des collectivités territoriales".

### Surgical procedure

Eleven ears have been implanted with a Carina T2 middle ear actuator (Cochlear Ltd., Sydney, Australia) at L'école de Chirurgie du Fer a Moulin (Paris, France). Implantations have been made for moderate to severe sensorineural hearing loss indications by coupling the actuator to the body of the incus bone. An incision was made superior to the ear canal before performing an atticotomy to expose the incus body. Using an actuator template, the fixation system is placed against the cortex in a position which would allow placing the actuator tip in contact with the incus body following the manufacturer's instructions for use [2]. Once the correct position is obtained, the actuator template is removed and the fixation system is put in place using four surgical screws. Afterwards, the TB was harvested en bloc and stored refrigerated for approximately 14 hours at -4 ˚C.

The day after implantation, the TBs were transported to Bicétre University Hospital (Le Kremlin-Bicétre, France) where the actuators are delicately mounted into the fixation system and loaded to the incus body (Fig 1) using the Cochlear™Carina® interface, the intraoperative test system provided by the manufacturer. The loading or contact of the actuator is measured by performing impedance measurements which are visualized in the accompanying fitting software. The impedance curve before and after loading are saved for further processing.

### Experimental procedure

Prior to exposing the TBs to the static magnetic field of the MRI scanner, each TB is scanned using a Siemens Definition AS CT scanner (Siemens Healthineers, Erlangen, Germany) using the following parameters: source voltage 140 kV, current 220 mA, slice thickness 0.6 mm acquired every 0.3 mm, pitch 0.8 mm, in-plane pixel spacing 0.156 x 0.156 mm, exposure time 1 s, convolution kernel U75u, and 512 x 512 pixel matrix.

Each TB is then exposed to the 1.5 T static magnetic field of a Magnetom Area Scanner (Siemens Healthineers, Erlangen, Germany) similar to when a patient would enter the MRI suite as shown in Fig 2. The TB is then rotated and put onto the patient bed, simulating

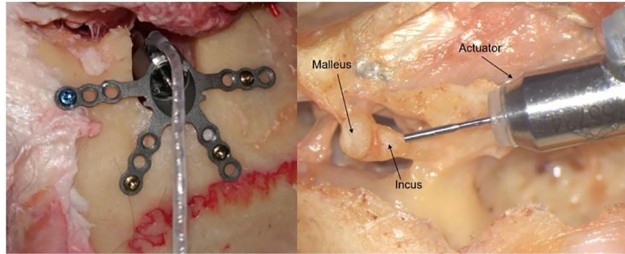

**Fig 1. Medial (left) and superior or middle fossa (right) view of the actuator loading.** The actuator is loaded by putting it in contact with the incus body. The point of contact is detected using the Cochlear™Carina® interface.

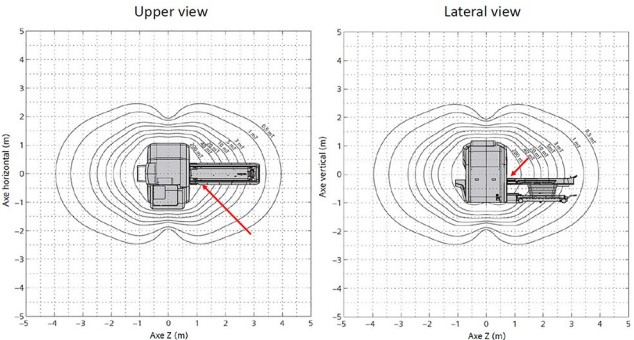

**Fig 2. Trajectory of entering the temporal bone (TB) in the magnetic field of the MRI suite on an upper view (left) and on a lateral view (right).** The temporal bone (TB) is exposed to the 1.5T static magnetic field of the MRI scanner similar to when a patient would enter the MRI suite. This figure shows the magnetic field distribution surrounding the scanner using isomagnetic field lines in black. The left image shows the upper view of the magnetic field and the right image shows the lateral view. The TB trajectory is illustrated in red.

when a patient would be lying down. The patient bed is then moved into the scanner at a velocity of approximately 30 cm/s, centering the TB in the scanner's isocentre where it is kept for 1 minute. The procedure is then reversed to remove the TB from the MRI suite and static magnetic field.

Inside the MRI control room and outside the static magnetic field, the actuator is connected to the Carina® interface in order to check the actuator loading via an electrical impedance measurement. Impedance curves were measured and saved after each exposure. When no resonance peak was observed it was retracted to check the functionality before reloading it.

This procedure has been repeated ten times per TB before re-scanning the TB in the CT-scanner.

## Data processing

**Analysis of CT images.** CT images obtained before and after exposing the TBs to the magnetic field were used to check the position of the tip of the actuator with respect to the ossicular chain. This position was analyzed using the distance between the edge of the ossicular chain (head of the malleus or body of the incus) along the longitudinal axis of the actuator and the connection point with the tip of the actuator. The position of the actuator tip and the position of the measurement lines are verified in the three orthogonal planes: axial, coronal and sagittal of the CT scan images (Fig 3). For the statistical analysis, we compared the measurement (length in mm) before and after MRI using a "Wilcoxon matched-pairs signed rank test" for each TB with the software Prism (GraphPad Software, Inc.).

**Analysis of impedance curves.** The loading of the actuator is checked during the experiment using the impedance measurement data acquired with the manufacturer's software. According to the surgical instructions [2] a decrease of $\geq 50\Omega$ at the resonance frequency indicates initial contact with the ossicles, after which the MicroDrive™ shall be turned an additional quarter turn to further load the actuator. This additional quarter turn will advance the actuator approximately 62.5 μm and ensure a good coupling efficiency [16, 17].

Impedance curves acquired after each exposure to the static magnetic field are compared to the impedance curve acquired after initial loading of the actuator. Impedance values are compared between both curves near the actuator's resonance frequency. When a difference of

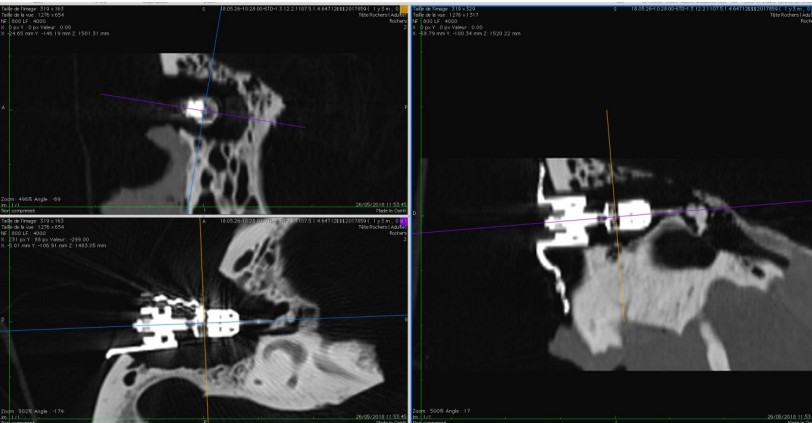

**Fig 3. Analysis of CT images.** In each available CT dataset, an axis was defined along the longitudinal axis of the actuator housing, shown in purple in the top left image and in blue on the bottom left figure. The intersection point between this axis and the malleus head was measured (right) and compared between images acquired before/after the experiment.

$\geq 50\Omega$ is observed, the actuator is considered to be unloaded. Impedance curves for one representative actuator after 1 and 10 exposures, both loaded and unloaded are shown in Fig 4.

## Results

### Actuator coupling as measured using CT images

The measured lengths (mm) between the actuator housing and the anterior edge of the malleus head are stated in Table 1 for each temporal bone. For three TBs the CT data was corrupted and no measurements could be made. No differences were observed in the measured length before and after exposure to the magnetic field (p = 0.265). Using the acquired images, it can be observed that the tip of the actuator is still on the same position related to the ossicles before and after the experiment. An illustration of the performed measurements is shown in Fig 5 for specimen 4093-L.

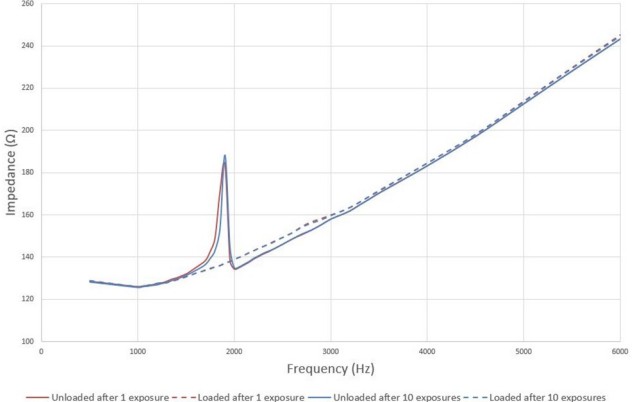

**Fig 4. Actuator impedance curves.** Impedance curves as measured using the Carina® interface are shown both for loaded (dotted) and unloaded (solid) actuators in a dotted and solid line respectively. Results after 1 exposure (red) and after 10 exposures (blue) to the magnetic field are shown.

**Table 1. Summary of results.**

| Specimen | Distance on CT before MRI (mm) | Distance on CT after MRI (mm) | Mean change in impedance (Ω, Min - Max) | Amount of MRI-induced unloadings |
|---|---|---|---|---|
| 3918-L | 9.402 | 9.400 | 1.88 (0.97 - 2.27) | 0 |
| 3918-R | 9.684 | 9.691 | 7.49 (5.51 - 8.75) | 0 |
| 4000-L | - | - | 3.22 (0.92 - 4.38) | 0 |
| 4000-R | 10.002 | 10.004 | 2.24 (1.30 - 2.92) | 0 |
| 4093-L | 9.949 | 9.951 | 5.25 (4.54 - 6.16) | 0 |
| 4093-R | 9.950 | 9.951 | 7.62 (6.81 - 8.43) | 0 |
| 4100-L | - | - | 2.75 (1.97 - 3.59) | 0 |
| 4100-R | - | - | 6.73 (6.12 - 7.45) | 0 |
| 4107-L | 10.030 | 10.040 | 5.77 (5.06 - 6.48) | 0 |
| 4107-R | 10.003 | 10.011 | 5.92 (4.21 - 7.13) | 0 |
| X-R | 10.003 | 10.013 | 6.78 (6.48 - 7.06) | 0 |

Summary of the obtained results per TB showing the distance between the edge of the ossicular chain and the entry point of the tip into the actuator body before (1st column) and after 10 exposures (2nd column) of the TB. Changes in impedance before and after scanning and MRI-induced unloadings are shown in the third and fourth column, respectively. Specimen are annotated by their anonymized reference, including a reference to the implanted ear (L: left, R: right). CT data was corrupted for 3 TBs, making it impossible to measure the distance.

## Actuator coupling as measured using the Carina interfacea TLA

A total of eleven Carina actuators have been implanted in TBs before being exposed ten times to the static 1.5 T magnetic field of an MRI scanner. The third column of Table 1 illustrates a summary of the results obtained using the impedance measurements. Differences in impedances at the resonance frequency between the initial loaded curve and each post-exposure measurement are shown for each specimen as the mean value over all exposures with the

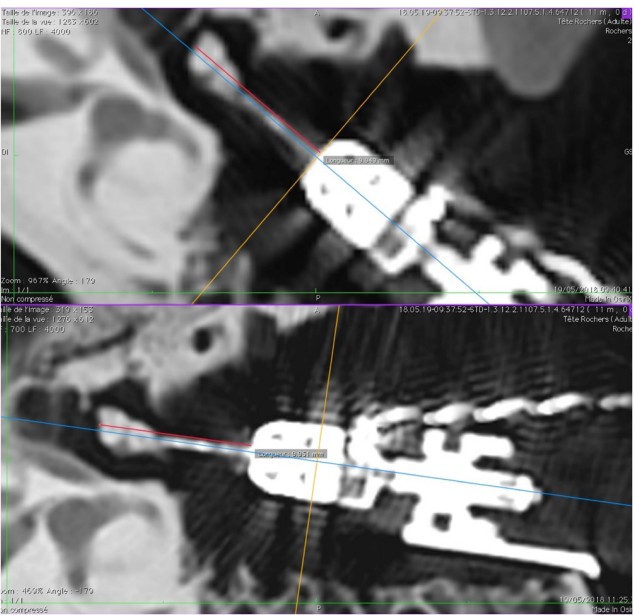

**Fig 5. Measurements on CT-scan before (9.949 mm, top) and after (9.951 mm, bottom) MRI on specimen 4093-L.**

minimum-maximum range between brackets. A visual representation of the obtained data per specimen is shown in Fig 6 below. For each specimen a boxplot is shown that visualizes the distribution of measured impedance differences after each exposure. The unloading threshold of $\geq 50\Omega$ that is specified in the manufacturer guidelines is highlighted with a dotted red line. As shown, no dislocations have been observed. No statistical analyses have been performed on the data as not a single dislocation has been observed.

## Discussion

Magnetically-induced forces are a significant safety hazard for patients with hearing implants, as most of these devices contain permanent magnets that can migrate during an MRI examination [9, 10]. For patients with cochlear or auditory brainstem implants, this risk is limited to a dislocation of the retention magnet implanted below the patient's skin. For patients with middle ear implants containing electromagnetic actuators, there is an additional increased risk of migration of the implanted actuator that may possibly damage the recipient's ossicular chain or functioning inner ear. Prior work by Todt et al. [12, 13] and Jesacher et al. [14] has indicated that this was a severe risk for patient's implanted with the previous generation of the MED EL Vibrant Soundbridge® implant.

The present study has provided more insights on how severe the risk of actuator migration could be for patient's implanted with a Cochlear™Carina® fully-implantable AMEI. Even though this implant is MRI contraindicated [2], the obtained results provide more insights in the matter which could be very useful when a Carina® patient has to undergo an emergency MRI examination, and for future generations of this device.

Experiments were performed by implanting eleven temporal bones with the electromagnetic T2 actuator coupled to the short process of the incus. This surgical position was preferred as it resembles the majority of reported surgical approaches in Carina® patients [1, 3–5]. Each temporal bone was entered and exited into the MRI suite in a clinically relevant way to simulate how a patient would enter the MRI suite during an examination. By crossing the scanner's fringe field, the actuators are exposed to a varying magnetic field. This will induce Lorentz forces on magnetized components according to [8]:

$$F_{trans} = \vec{\mu_m} \cdot \vec{\nabla} B \qquad (1)$$

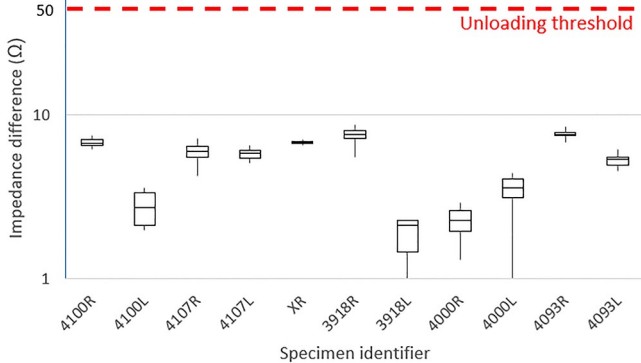

**Fig 6. Summary of impedance differences at the actuator resonance frequency before and after exposure.** For each specimen, a box plot is shown that indicates the spread in impedance difference measured after each exposure. The impedance difference of $\geq 50\Omega$ that is considered to be the unloading threshold based on manufacturer guidelines is indicated with a dotted red line. Note that the y-axis is organized logarithmically to allow a clear visualization.

with $\vec{\mu_m}$ the magnetization vector and $\vec{\nabla}B$ the magnetic field gradient. This force will be maximal at the maximum $\vec{\nabla}B$, which will be at the edge of the scanner bore at the entrance of the scanner. The presented experiments were however limited to the clinically relevant case, as it is highly unlikely that a patient will cross the fringe field this close to the edge of the scanner bore. Once the actuators were put onto the patient bed, the bed was moved into the scanner bore up to the point where the actuators were in the scanner isocenter. In the scanner's isocenter, the homogeneity of the magnetic field will be maximal, and so will be the induced torque on magnetized components, according to [8]:

$$\vec{\tau} = \vec{\mu_m} \times \vec{B} \qquad (2)$$

By combining traversing the fringe field in a clinically relevant manner and putting the actuators in the scanner isocenter in a single experiment, the actuators were exposed to realistic forces and torques that may occur during scanning. Forces and torques due to the dynamic gradient field are not taken into account due to the significantly lower magnitude of this field compared to the scanner's static magnetic field.

In order to determine if the actuator has migrated due to the induced forces and torques, two measurements were used. By comparing CT images acquired before and after the experiment, positional changes larger than the scanner resolution of 0.30.6 mm can be observed. To increase the resolution of this method, electrical impedance curves are measured which can be used to detect if the actuator remained loaded onto the short process of the incus. Using the CT data, no actuator position changes could be observed in any of the temporal bones. By comparing the impedance of the actuator before and after loading, any dislocations could be detected. The threshold for actuator dislocation was a difference of 50 Ω at the actuator resonance, which has not been observed in any of the temporal bones.

It should be noted however that the presented methods are only capable of detecting permanent position changes of the actuator. The presented study therefore only focuses on the coupling stability of the T2 actuator when exposed to a 1.5 T MRI magnetic field, and it does not study any transient vibrational or positional changes that could cause patient discomfort or unintended acoustic stimulation of the patient. As sound can be picked up at very low intensities, MRI induced actuator vibrations of all acoustic hearing implant should be investigated in further study.

During this study, eleven implanted temporal bones have been exposed over a total of 110 times to the static magnetic field of a 1.5T MRI scanner. Although not a single position change has been observed in this large amount of exposures, care should always be taken to avoid exposing patients with electromagnetic AMEI to high gradient magnetic fields. The presented research did not identify any adverse events for the patients, but that does not imply patient safety [18, 19]. The presented work indicates that the risk of actuator dislocation for the Cochlear Carina middle ear implant system due to magnetically induced forces and torques is low. The Cochlear™Carina® system is MRI contraindicated, and more experiments would be necessary to demonstrate the safety of the full system from a regulatory perspective.

Despite its MRI-unsafe label, it could be required to expose a patient with a Carina implant to a 1.5T MRI examination in life threatening conditions. The presented research indicates that the Carina® interface can be used as a post-scan check to verify actuator coupling.

## Conclusion

The presented work has provided more insights regarding the risk of dislocating the Carina middle ear actuator due to MRI-induced displacement forces and torques. Temporal bones have been implanted with Cochlear Carina middle ear actuators coupled to the incus short

process. Two measurement modalities have been used to investigate any permanent actuator displacement due to these forces or torques. Dislocations of the actuator have not been observed during the analysis of the obtained dataActuator dislocation or positional changes have not been observed in any of the presented experiments, indicating that the risk on dislocating the Carina middle ear actuator when coupled to the incus short process is limited.

## Supporting information

**S1 File.**
(XLSX)

## Acknowledgments

The authors thank Rish Verma and Antonin Rambault for providing technical support during the experiments.

## Author Contributions

**Conceptualization:** Guy Fierens, Jérôme Nevoux.

**Data curation:** Guy Fierens.

**Formal analysis:** Guy Fierens, Jérôme Nevoux.

**Funding acquisition:** Jérôme Nevoux.

**Investigation:** Guy Fierens, Jérôme Nevoux.

**Methodology:** Guy Fierens, Jérôme Nevoux.

**Project administration:** Jérôme Nevoux.

**Resources:** Farida Benoudiba, Jérôme Nevoux.

**Supervision:** Nicolas Verhaert, Jérôme Nevoux.

**Validation:** Guy Fierens, Jérôme Nevoux.

**Writing – original draft:** Guy Fierens, Jérôme Nevoux.

**Writing – review & editing:** Guy Fierens, Nicolas Verhaert, Farida Benoudiba, Marie-France Bellin, Dennis Ducreux, Jean-François Papon, Jérôme Nevoux.

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
