## [Decision Letter · Decision Letter 0]

20 Jan 2020

PONE-D-19-30029

Stability of the standard incus coupling of the Carina middle ear actuator after 1.5T MRI

PLOS ONE

Dear Mr. Fierens,

Thank you for submitting your manuscript to PLOS ONE. After careful consideration, we feel that it has merit but does not fully meet PLOS ONE’s publication criteria as it currently stands. Therefore, we invite you to submit a revised version of the manuscript that addresses the points raised during the review process.

Additional Editor Comments:

Dear authors,

The decision for your manuscript is "minor revision". Please address the comments made by the reviewers.

Also, I have an additional question: The authors concluded that "the risk on dislocating the Carina middle ear actuator when coupled to the incus short process is limited" - Could the authors please provide more information on how this conclusion was reached?

Best regards,

We would appreciate receiving your revised manuscript by Mar 05 2020 11:59PM. To enhance the reproducibility of your results, we recommend that if applicable you deposit your laboratory protocols in protocols.io, where a protocol can be assigned its own identifier (DOI) such that it can be cited independently in the future. For instructions see: http://journals.plos.org/plosone/s/submission-guidelines#loc-laboratory-protocols

We look forward to receiving your revised manuscript.

Kind regards,

Rafael da Costa Monsanto, M.D.

Academic Editor

PLOS ONE

Journal Requirements:

2. Please disclose the funding from Cochlear Ltd. in your competing interests statement.

3. In your data availability statement you write, that all relevant data are within the paper. Please ensure you have provided the individual data points used to create the figures and determine means, medians and variance measures presented in the results, tables and figures (http://journals.plos.org/plosone/s/data-availability#loc-faqs-for-data-policy). If these data cannot be publicly deposited or included in the supporting information, e.g. due to patient privacy or ownership by a third party, explain why and explain how researchers may access them.

Reviewers' comments:

Reviewer's Responses to Questions

**Comments to the Author**

1. Is the manuscript technically sound, and do the data support the conclusions?

Reviewer #1: Partly

Reviewer #2: Yes

Reviewer #3: Yes

2. Has the statistical analysis been performed appropriately and rigorously? 

Reviewer #1: Yes

Reviewer #2: Yes

Reviewer #3: Yes

3. Have the authors made all data underlying the findings in their manuscript fully available?

Reviewer #1: Yes

Reviewer #2: Yes

Reviewer #3: Yes

4. Is the manuscript presented in an intelligible fashion and written in standard English?

Reviewer #1: Yes

Reviewer #2: Yes

Reviewer #3: Yes

5. Review Comments to the Author

Reviewer #1: I read the manuscript entitled “Stability of the standard incus coupling of the Carina middle ear actuator after 1.5T MRI” with great interest. In this study, the authors aimed to measure the coupling stability of an actuator of a fully implantable AMEI after MRI scans. The subject of the study is clinically relevant, and the results are presented in an intelligible fashion.

Although I commend the authors for tackling such an important subject, I do have a few concerns regarding the methodology that the authors could perhaps clarify.

1. For stapedotomy, the angle between the prosthesis used and the incus long process is known to be ideal at 90 degrees (https://www.ncbi.nlm.nih.gov/pubmed/12671422). I wonder that there must be perfect alignment between the actuator and the ossicular chain for optimal device performance. Why did the authors not perform these angle measurements?

2. The data from 3 TB CT scans was corrupted. Therefore, only 8 CT scans could be analyzed. I think this is a small sample to get further conclusions.

3. What does LTA mean?

4. I believe that statistical analysis for mean impedance values (before and after MRI) are necessary.

Reviewer #2: Dear Author, your manuscript is well done and interesting. It is the first that evaluate the compatibility of the Carina Implant with the MRI. Your job is ready for the publication but one defect is that only the trasductor is studied. Can you evaluate all the device? You can stabilized implant and microphone with screws and then do the MRI. Please, let me know your opinion about this possibility.

Reviewer #3: The authors of the present study conducted an extremely pertinent analysis of the subject in question. They clearly introduced the importance of magnetic resonance imaging in the growing world of implantable hearing aids. The study was conducted with fidelity of technical application, respecting the necessary steps to conduct the imaging exams in order to minimize the bias of selection or variation of radiological technique. The procedure of implantation of the anatomical specimens respected the indication criteria according to the latest manufacturer's guidelines. Actuator T2 dislocations were analyzed not only according to the computed tomography exam, which could lead us to a technical bias, but also according to the average of the impedance curves before and after 10 exams. MRI scan using the manufacturer's TLA. Finally, the authors concluded that and according to the results presented, little effect of the factor (magnetic resonance) on the implant analyzed, when using the classic anvil short coupling method.

6. PLOS authors have the option to publish the peer review history of their article (what does this mean?). If published, this will include your full peer review and any attached files.

Reviewer #1: Yes: Henrique Furlan Pauna, MD

Reviewer #2: Yes: Luca Bruschini

Reviewer #3: Yes: IULO SERGIO BARAUNA FILHO

---

## [Author Response · Author response to Decision Letter 0]

10 Mar 2020

Dear reviewers,

We thank you for your time and effort to review our submission. We have addressed your comments in detail in the attached rebuttal letter.

Kind regards,

Guy Fierens

---

## [Decision Letter · Decision Letter 1]

19 Mar 2020

Stability of the standard incus coupling of the Carina middle ear actuator after 1.5T MRI

PONE-D-19-30029R1

Dear Dr. Fierens,

We are pleased to inform you that your manuscript has been judged scientifically suitable for publication and will be formally accepted for publication once it complies with all outstanding technical requirements.

With kind regards,

Rafael da Costa Monsanto, M.D.

Academic Editor

PLOS ONE

Additional Editor Comments (optional):

Reviewers' comments:

Reviewer's Responses to Questions

**Comments to the Author**

1. If the authors have adequately addressed your comments raised in a previous round of review and you feel that this manuscript is now acceptable for publication, you may indicate that here to bypass the “Comments to the Author” section, enter your conflict of interest statement in the “Confidential to Editor” section, and submit your "Accept" recommendation.

Reviewer #1: All comments have been addressed

Reviewer #2: All comments have been addressed

2. Is the manuscript technically sound, and do the data support the conclusions?

Reviewer #1: Yes

Reviewer #2: Yes

3. Has the statistical analysis been performed appropriately and rigorously? 

Reviewer #1: Yes

Reviewer #2: Yes

4. Have the authors made all data underlying the findings in their manuscript fully available?

Reviewer #1: Yes

Reviewer #2: Yes

5. Is the manuscript presented in an intelligible fashion and written in standard English?

Reviewer #1: Yes

Reviewer #2: Yes

6. Review Comments to the Author

Reviewer #1: I believe the authors have properly addressed all topics regarding their submission. I believe it is ready for publication.

Reviewer #2: This manuscript describ the stability of the Carina transducer during a MRI. It could be the first step to demonstrate that a patient with the Carina implant can perform the MRI. This manuscrip is well done and ready for the publication

7. PLOS authors have the option to publish the peer review history of their article (what does this mean?). If published, this will include your full peer review and any attached files.

Reviewer #1: Yes: Henrique Furlan Pauna

Reviewer #2: Yes: Luca Bruschini

---

## [Editor Report · Acceptance letter]

23 Mar 2020

PONE-D-19-30029R1 

Stability of the standard incus coupling of the Carina middle ear actuator after 1.5T MRI 

Dear Dr. Fierens:

I am pleased to inform you that your manuscript has been deemed suitable for publication in PLOS ONE. Congratulations! Your manuscript is now with our production department. 

With kind regards,

on behalf of

Dr. Rafael da Costa Monsanto 

Academic Editor

PLOS ONE